# Long-Horizon Dialogue Understanding
# for Role Identification in the Game of Avalon with Large Language Models

**Simon Stepputtis[1], Joseph Campbell[1], Yaqi Xie[1], Zhengyang Qi[1], Wenxin Sharon Zhang[1],**
**Ruiyi Wang[1], Sanketh Rangreji[1], Charles Michael Lewis[2], Katia P. Sycara[1]**
[1]Carnegie Mellon University, [2]University of Pittsburgh
{stepputtis, jcampbell, yaqixie}@cmu.edu, ml@sis.pitt.edu
{zqi2, wenxinz3, ruiyiwan, srangrej, sycara}@andrew.cmu.edu

## Abstract

Deception and persuasion play a critical role in long-horizon dialogues between multiple parties, especially when the interests, goals, and motivations of the participants are not aligned. Such complex tasks pose challenges for current Large Language Models (LLM) as deception and persuasion can easily mislead them, especially in long-horizon multi-party dialogues. To this end, we explore the game of *Avalon: The Resistance*, a social deduction game in which players must determine each other's hidden identities to complete their team's objective. We introduce an online testbed and a dataset containing 20 carefully collected and labeled games among human players that exhibit long-horizon deception in a cooperative-competitive setting. We discuss the capabilities of LLMs to utilize deceptive long-horizon conversations between six human players to determine each player's goal and motivation. Particularly, we discuss the multimodal integration of the chat between the players and the game's state that grounds the conversation, providing further insights into the true player identities. We find that even current state-of-the-art LLMs do not reach human performance, making our dataset a compelling benchmark to investigate the decision-making and language-processing capabilities of LLMs. Our dataset and online testbed can be found at our project website: https://sstepput.github.io/Avalon-NLU/

## 1 Introduction

Despite the remarkable progress of large language models (LLMs) in natural language understanding and generation, they have largely been applied and evaluated in question-answering (Brown et al., 2020), instruction following (Driess et al., 2023), and cooperative dialogue (Madotto et al., 2020) tasks. An oft-overlooked, yet important setting is that of multi-party dialogue in cooperative-competitive scenarios, where participants hold pri-

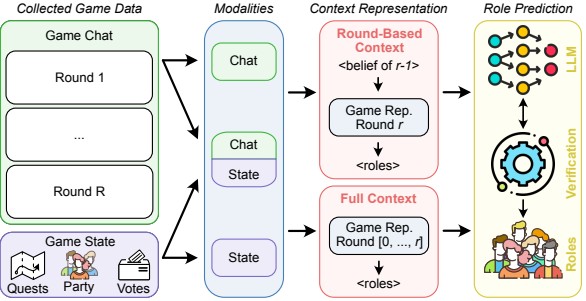

Figure 1: Schematic representation of the *Avalon* role prediction pipeline. We experiment with three distinct modalities: Chat only, Chat and State, and State only. These representations are provided to the Language Model (LLM) either by using data from a single round complemented with a carried-over belief (round-based context) from the preceding round or by using the entire history since the game's beginning (full context). Subsequent role predictions made by the LLM are validated for consistency with the predefined *Avalon* roles.

vate and – possibly competing – beliefs and agendas, yet seek to cooperate in service of a shared goal. While humans excel at such tasks and are capable of reaching group consensus even in the presence of bad faith actors and deception, there are significant challenges inherent to the setting which state-of-the-art LLMs are ill-suited to handle (Wang et al., 2023b).

In this work, we aim to explore these limitations by introducing a new benchmark and associated dataset for a multi-party cooperative-competitive task based on the game of *Avalon: The Resistance*[1]. Our task features two teams of players with hidden roles and conflicting goals; one side seeks to fulfill a series of objectives and the other sabotage them. Through rounds of dialogue and voting, the players must discover each other's identity while concealing their own to win. We model the identification of player roles as a *long-horizon* dialogue understanding task, and show that it poses a challenge for

---

[1]A game by Don Eskridge: https://en.wikipedia.org/wiki/The_Resistance_(game)

recent state-of-the-art LLMs due to three factors: 1) a large number of participants in a multi-party dialogue, 2) the need to ground reasoning in both dialogue and game state, and 3) the active usage of deception and persuasion by players.

While prior works have explored dialogue understanding in multi-party settings, they typically involve a small number of participants (Zahiri and Choi, 2017) or unnatural dialogue captured from online forums (Lowe et al., 2015). Our task, in contrast, involves natural dialogues with *six* participants, each of whom have their own privileged information and agendas. This significantly increases the complexity of the task, as participants may address each other over longer horizons. Compounding the issue is the need to incorporate game state into the reasoning process, further increasing the horizon if translated to natural language as is common in zero-shot paradigms. However, the ability to reason over extended horizons is crucial in our context, as inconsistencies in behavior, evasive responses, and self-contradictions which can be used to identify deceptive behavior may only manifest over time.

We explore a selection of recent, publicly available LLMs, varying in size and training data, to evaluate their effectiveness in understanding long-horizon relations in our proposed benchmark. Under the assumption that LLMs encode large amounts of general knowledge, we aim to ascertain whether specific state representations can facilitate the comprehension of long-horizon tasks. We find this to be a critical aspect given the small context windows associated with many state-of-the-art LLMs, despite recent improvements (Dao et al., 2022; Ainslie et al., 2023; Press et al., 2021; Packer et al., 2023).

In addition, we publicly release our dataset – as well as our browser-based version of Avalon – in order to promote further research in this direction. Consisting of 20 games with 30 unique human players and 19 unique team compositions, our dataset comprises 2384 pieces of dialog with hand-annotated persuasion strategies for each player, deception strategies for evil players, player beliefs over the roles of other players throughout the game, and ground truth game state. To the best of our knowledge, this is the only high-quality dataset of its kind for a *long-horizon* multi-party dialogue featuring deception and persuasion. In keeping with recent trends advocating for quality over quan-

tity (Gunasekar et al., 2023; Raffel et al., 2020; Longpre et al., 2023), we expect this dataset to be useful for both fine-tuning and evaluation of models. During data collection, we have ensured that no spurious information channel exists between the players, ensuring that all conversation relevant to the game is captured by our dataset. Concretely, our contributions are as follows:

- A testbed and dataset containing 2384 utterances from 20 human player games hand-annotated with strategies (persuasion and deception), player beliefs, and game state.

- A comprehensive analysis of LLM performance in our proposed multimodal long-horizon dialogue understanding benchmark, including persuasive and deceptive behavior.

- An exploration of the limitations of current models and the introduction of state representations that can improve long-horizon dialogue modeling.

## 2 Related Work

**Dialogue Understanding in Games:** Dialogue understanding tasks such as the identification of intentions and motivations are essential for successful conversation (Weld et al., 2022), and are associated with broader human cognitive activities. While games have long served as challenges in the AI community, they often lack such dialog understanding due to the difficulty in generating and evaluating realistic dialog in commonly employed reinforcement learning paradigms (Vinyals et al., 2019). As a result, cooperative games with human partners often utilize simple, codified communication protocols such as in the game of *Hanabi* (Siu et al., 2021) or ignored entirely (Serrino et al., 2019). When dialogue is incorporated (Zhao and Eskenazi, 2016; Pang and Wang, 2020), it is often through dialogue state tracking (DST) (Ren et al., 2018) in which participant beliefs (Oguntola et al., 2023) and intentions are modeled as semantic *slots* (Lee et al., 2021), and are inferred throughout the course of a conversation. However, cooperative-competitive games involving negotiation, persuasion, or deception often require a more nuanced application of language that leaves room for subjectivity and interpretation. Past work on strategic dialogue systems has avoided these issues by focusing on simpler settings (Lewis et al., 2017; Keizer et al., 2017; Wang

et al., 2019), which involve only a single partner, shorter dialogue contexts, or simpler strategies.

Recent advances in LLMs have shown great potential (Liu et al., 2023; Li et al., 2023) across varying problems, including conversing (OpenAI), knowledge parsing (Jiang et al., 2023; Zhang et al., 2023), and instruction following (Alayrac et al., 2022; Stepputtis et al., 2020; Xie et al., 2023), leading to the development of capable language-based agents (Team et al., 2021; Wang et al., 2023a; Driess et al., 2023). As shown in White et al. (2023), the inherent knowledge embedded in these models' weights allows for intricate answers, if the prompt is posed correctly. Such agents have recently been successfully applied to the game of *Diplomacy* (FAIR et al., 2022; Bakhtin et al., 2022), although a significant source of relevant domain data was required for fine-tuning. This differs from the game of *Avalon*, in which a) all communication is public and b) hidden roles explicitly encourage deception, which leads to a more challenging dialogue understanding problem as an agent must carefully understand, analyze, and ground each player's utterance.

**Deception and Persuasion in Dialogue:** While deception is increasingly studied in terms of misinformation on social media (Shu et al., 2017), in this work we focus on its analysis and detection in dialogue. Unlike prior works which explore deception through analysis of verbal (Hirschberg et al., 2005) or visual (Soldner et al., 2019) cues in spoken language from two-party dialogues, our benchmark is based on textual linguistic cues in multi-party dialogues. Although datasets have previously been introduced for the games of *Mafia* (Ibraheem et al., 2022) and *One Night Werewolf* (Lai et al., 2022), we find *Avalon* to be a significantly more challenging task due to the increased game length, resulting in more than double the number of utterances per game in our dataset – 49, 64, and 119 for *Mafia*, *Werewolf*, and *Avalon*, respectively. This requires dialogue models to reason over significantly longer context horizons, but also provides enough information for us to reason over hidden player roles as opposed to simply inferring utterance labels. In addition, we provide high-quality annotated persuasion strategy labels for each utterance in our dataset, following annotation schemes introduced in prior works (Yang et al., 2019; Lai et al., 2022), as well as deception labels, covering the type of lies utilized by the evil players (Houston et al., 2013).

Though persuasion has often been studied in the context of negotiation (Lewis et al., 2017; Keizer et al., 2017) or other two-party dialogues (Wang et al., 2019), multi-party persuasion analysis is often limited to online social media discussions (Althoff et al., 2014; Tan et al., 2016) which often meaningfully differ from real-time dialogue.

## 3 The Game of Avalon

In this section, we provide a description of the game *Avalon: The Resistance*[2], which is a social deduction board game that can be played by five to ten players. Players assume various roles in the game that differ in knowledge and goals, including the Assassin, Merlin, Morgana, and Percival. *Avalon* is a cooperative-competitive game in which two groups of players attempt to infer the roles of the other players by forming allegiances while hiding or intentionally pretending to be a role different from their own. In this work, Avalon is played among six human players $\mathcal{P} = \{p_1, \ldots, p_6\}$ – four good, two evil. Generally, the goal of the evil players is to hide their identity and to convince the good players that they can be trusted.

The game progresses through up to five rounds – referred to as quests – with players engaging in discussions and voting to determine the composition of a party that is subsequently sent on a quest. The required size of a party that is sent on a quest differs in each round and has to be approved by a public majority vote amongst all players. After a party has been formed, each player in that party votes anonymously whether or not the quest should succeed, requiring all players to vote for success in order to succeed the quest. If the evil players succeed in failing three quests, they automatically win the game. Similarly, if the good players succeed three quests, they win the game, unless the Assassin can identify who plays Merlin (see Section A.1). If the Assassin is successful, evil wins the game instead.

*Avalon: The Resistance* introduces multiple special roles that impact gameplay, adding layers of strategic depth to the game's dynamics. In our setting, the roles consist of Merlin, Percival, and two Loyal Servants as the forces of good, with Morgana and the Assassin on the side of evil. Detailed role descriptions can be found in Section A.1

We further impose additional rules to facilitate

---

[2]The game's rules can be found here: `https://www.ultraboardgames.com/avalon/game-rules.php`

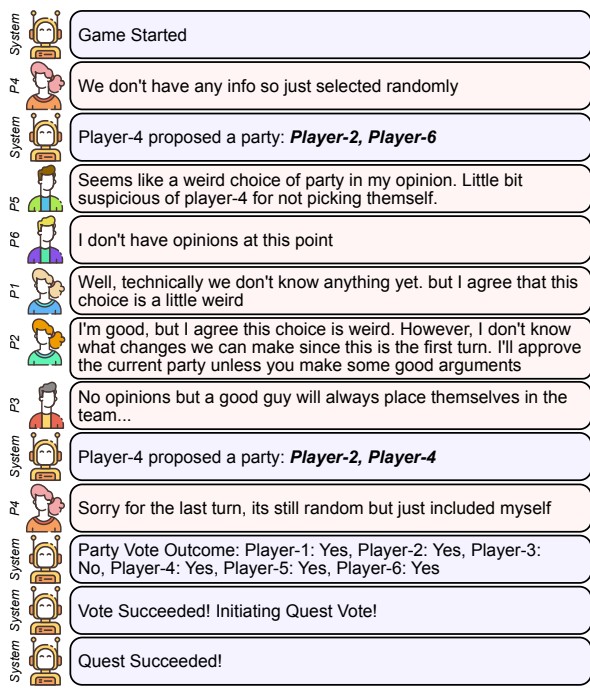

Figure 2: Example of the players' conversation during round one. Player *P4* is the quest leader.

the online nature of our data collection. Particularly, we enforce a turn-based discussion in which each player has a fixed amount of time to convey their thoughts via a chat interface, thus ensuring that all game-related interactions are captured. Should a player exceed their allotted time, the game will automatically progress to the next player, initiate appropriate votes, or apply default votes depending on the situation. More details can be found in section A.2

## 4 Dialogue Understanding

We formulate our problem as identifying a player's character $c_i$ given the chat history $h_{\text{chat}}$ and game state $h_{\text{state}}$. Formally, predict the role $P_{\text{role}}(c_i|h_{\text{chat}}, h_{\text{state}})$ for each player in $\mathcal{P}$. Additionally, we also predict which of the six players is Merlin: $P_{\text{Merlin}}(p_{\text{Merlin}}|h_{\text{chat}}, h_{\text{state}}, e)$ where $e$ is the privileged knowledge of the evil players.

In the remainder of this section, we address the two fundamental issues of 1) understanding long-horizon conversations, and 2) utilizing the environment's state to enhance the inference capabilities of the LLM.

### 4.1 Long-Horizon Dialogue Representation

Long-horizon conversations between multiple participants pose challenges for LLMs as their capability for comprehending and tracking context is limited. Especially in situations in which deception behavior is present, as in *Avalon*, identifying such behavior hinges on multiple factors. For example, consistency when voting for parties depending on which players are on it, as well as identifying inconsistencies in a player's arguments. These serve as an indicator for deceptive behavior, and consequently, the true nature of the player's character $c$. We propose to break up conversations into "rounds" $r$ representing a single round of discussion in which each player has had the chance to speak at least once. Breakpoints are introduced at the current quest leader's position. An example round can be seen in Figure 2. In this example, we propose that players' roles can be predicted given only the current round $r_i$ and a structured belief state $b$ about each player's identity that carries over between rounds. To track the identities of players across rounds, the model is provided with a list of initial player beliefs $b = [b_1, \ldots, b_6]$, where each belief $b_i \in \{\text{good}, \text{evil}, \text{merlin}, \text{unknown}\}$, as context. The context is provided prior to consuming the next round of conversation. Finally, the model is tasked to provide an updated list of player beliefs $b_{\text{next}}$ after each round, which subsequently serves as an input for the next round $r_{t+1}$.

Figure 1 describes the three modalities we are building from the game's chat and game state. Each of these modalities can be used in either context representation. Particularly, the game is either conveyed to the LLM as the complete history – which we refer to as full context – from the beginning of the game up to the point of evaluation or as a round with a carried-over belief $b$ holding the role predictions after the previous round, which we refer to as round-based context. The respective context is then utilized by the LLM through TypeScript (see Section 4.3.1), allowing for validation of the generated response, ensuring that a valid role has been assigned to each player.

### 4.2 State Representation

Recent LLMs have shown impressive, human-like, conversational capabilities and are increasingly deployed as public-facing conversational agents (OpenAI). However, the shortcoming of these models is that their interactions are mostly cooperative and they assume that language does not need to be grounded in the world's state. With our testbed and accompanying dataset, we provide an intriguing, yet simple world in which language can be

grounded in a structured state representation of the relevant environment.

In the game of *Avalon*, the relevant state information includes the currently proposed party, the record of successful and failed quests, the players that have been part of parties in prior rounds, as well as previous party formation votes. At its core, we convert state information into linguistic statements that are given to each LLM prior to the player's conversations. By utilizing this technique, the LLM can leverage its inherent zero-shot and few-shot capabilities to solve our task of identifying hidden player roles by interleaving specially designed inputs, with little to no task-specific fine-tuning. In *Avalon*, states are conveyed to the LLMs in two manners: 1) through a global game state that incorporates past quest outcomes, and 2) state changes that happen in the currently evaluated round.

### 4.2.1 Global State

The game's global state is converted into text by utilizing a set of templates. Most importantly, the outcome, as well as parties and votes for prior quests are translated as follows:

- quest-$i$: $o_i$ (party: $\boldsymbol{p}_j$ | player votes: $\boldsymbol{v}_j$)

where $i$ is quest $q_i$'s ID, $o_i \in \{\text{success}, \text{failure}\}$ is the outcome of quest, $\boldsymbol{p}_j$ is the list of players that went on quest $q_i$ and $\boldsymbol{v}_j$ is a list of tuples including the vote from each player of whether or not they approved of party $\boldsymbol{p}_j$. For example, the game's state after the first successful quest conducted by player one and two could look as follows: *"quest-1: success (party: player-1, player-2 | player votes: player-1: yes, player-2: yes, player-3: yes, player-4: yes, player-5: yes, player-6: yes)"*. Further, the global state also covers the currently proposed party $\boldsymbol{p}_{\text{proposed}}$ in order to provide context for the players' discussion.

### 4.2.2 Round-Based State Change

In addition to the game's global state each round of conversation can be accompanied by proposing parties, voting for such parties, and conducting quests. Such changes have an effect on the global state; however, the global state only represents the game prior to the currently ongoing discussion. Thus, changes that occur during a round are communicated through a 7-th player named "system". Figure 2 demonstrates the conversation between the six players during round one. Here, the system

---

**system:** You are a helpful assistant that uses the chat between six players, player-1 to player-6, who play Avalon: The Resistance (a cooperative-competitive game) to identify who is Merlin, Good or Evil. There are two evil players, which can usually be found because they are deceptive and lie about the good player's roles and vote for quests and parties irrationally. For Merlin, watch out for individuals with knowledge of evil players' identities, insightful comments beyond their role, and caution regarding mission teams or specific players.

**human:** The current state is: *<STATE>* (See Section 4.2.1) The current party proposal is: *<PARTY>* (See Section 4.2.1)
Your initial belief is: *<BELIEF>* (See Section 4.1)
This is the chat between player-1 to player-6:
*<CHAT>* (See Section 4.2.2)
What do you think is the role of each player? Please do not explain your answer, do not elaborate on it further, and do not say that these are just guesses; only provide the list and nothing else.

Table 1: Prompt used for role inference.

---

user provides updates about the game's global state during the round.

### 4.3 Tasks and Prompt Generation

The prompts used to predict roles $P_{\text{role}}$ and predict Merlin $P_{\text{Merlin}}$ have been hand-designed to covey the basics of the game as well as the expected output format. The prompts for our two prediction tasks are shown in Tables 1 and 4, respectively.

Each prompt is designed such that various information can be given dynamically to the models for inference. In particular, we are evaluating the impact of providing information about the game's state to the model. Prompts shown in Tables 1 and 4 demonstrate the case in which chat $\boldsymbol{h}_{\text{chat}}$ and state $\boldsymbol{h}_{\text{state}}$ information are given. If only chat is used, lines indicating information about the <STATE> and current <PARTY> are omitted. Further, messages from the 7-th "system" player are removed from the <CHAT> information, thus only retaining the players' utterances. Similarly, if only the game's state is desired, utterances from all players are removed from the <CHAT> information.

### 4.3.1 Structured LLM Outputs

While very powerful, LLMs struggle adhering to structured outputs, as, for example, required in our role-prediction task. An LLM tasked to produce a bullet-point list of player names followed by one of the roles results in varying answer qualities ranging from half-populated lists to free-form responses with no discernible list, making subsequent utilization of the predictions difficult. However, many LLMs exposed to code during training

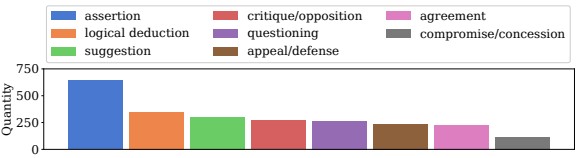

Figure 3: Distribution of persuasion strategies.

can generate JSON strings and understand simple code blocks. To capitalize on this feature, we utilize TypeChat (Microsoft, 2023), which, given a schema definition, tasks LLMs with answering all user queries with a JSON string that conforms to the provided schema. A benefit of this approach is that the LLM response can be verified to satisfy the desired response schema by utilizing the Type-Script compiler. In cases where the response is incomplete or otherwise insufficient, TypeScript's compiler error message can be used to formulate a follow-up query to the LLM to generate a valid response. We utilize this approach to generate role predictions for each player, as well as for predicting which player is Merlin. Our schema definitions can be found in Section C.3 and enforce that a valid role is predicted for each player. However, our schema does not enforce that; for example, the role of Merlin can only be assigned to a single player.

## 5 Experiments

The following sections detail our data collection process and provide comparisons between three current state-of-the-art LLMs, including fine-tuned versions of these models, in utilizing different modalities and context representations.

### 5.1 Data Collection

We created a browser-based version of *Avalon: The Resistance* that allows for unsupervised data collection from six participants. During the game, we collect the conversations between the players and the state of the game throughout each session. Further, we instructed participants to self-label persuasion strategies (see Section A.1.4 for details) for each of their chat messages, as well as indicating beliefs about the other players' roles throughout the game. In total, we collected 20 games over 24 hours of in-game time from a total of 30 unique participants, forming 19 different teams of six players. Participants were required to be at least 18 years old and familiar with the general rules of *Avalon*. The Office of Research Integrity and Compliance sanctions the experiment and data collection.

The dataset released with this paper contains over 2300 utterances and corresponding hand-labeled persuasion and deception strategies. Games take an average of 1 hour and 12 minutes, with the shortest game lasting only 18 minutes and the longest game over 3 hours. Seven of the 20 games were won by the good players, while 13 were won by evil – nine of which were won through the Assassin correctly identifying Merlin.

We split the dataset into 14 training and six testing games. The test games were selected such that they contain three good victories and three evil victories, where two of the three evil victories are through the assassin and one through successfully failing three quests. To form a human baseline, we created Google Forms surveys and asked players not involved in the test games to label the roles (see Section A.3) by reading over the recorded games from the perspective of an external observer.

#### 5.1.1 Data Processing

Due to the chat-based nature of our data collection process, the collected utterances are of reasonable quality. However, we clean the data with an automatic spell-checker that corrects any word with a Levenshtein Distance between an unknown word and an English target dictionary. Similarly, player names are corrected with a custom dictionary containing the correct spelling of the player's names. After this procedure, the remaining unknown words are corrected manually, which mostly contain abbreviations of long or unusual player names. After spelling and player names are corrected, the data is anonymized by replacing player names with "player-X" where X is the index of the player in each particular game. Note that this means that in two different games, player-1 can refer to a different person. This replacement was done to not bias the models towards certain players having identifiable play styles.

#### 5.1.2 Strategy Labels

Every utterance in our dataset is accompanied by a label indicating one of our eight different persuasion strategies (see Figure 3 and Section A.1.4). Further, if not truthful, utterances of evil players are labeled with an additional deception strategy label, covering the three common types of lies (Houston et al., 2013): commission, omission, and influence. Table 6 in Section D.2 shows the prediction capabilities of GPT-4, GPT-3-Turbo, and Llama-2-13b of our eight prediction strategies, including fine-

Table 2: F1 scores for role prediction and identifying Merlin given privileged knowledge. We report results using the round-based (first value) and full context (second value). Models are evaluated ten times on the six test games.

| | Model | Familiar | Trained | Modalities | | All-Role Prediction | | | Evil find Merlin | |
|---|---|---|---|---|---|---|---|---|---|---|
| | | | | Chat | State | Good | Evil | Merlin | Final | Anytime |
| 1 | Gpt-4 | ✓ | | ✓ | | 0.67 / 0.67 | 0.48 / 0.55 | 0.36 / 0.20 | 0.17 / 0.17 | 0.83 / 0.67 |
| 2 | Gpt-4 | ✓ | | | ✓ | 0.59 / 0.57 | 0.20 / 0.33 | 0.06 / 0.31 | 0.17 / 0.33 | 0.17 / 0.50 |
| 3 | Gpt-4 | ✓ | | ✓ | ✓ | 0.67 / 0.68 | 0.46 / 0.58 | 0.05 / 0.27 | 0.00 / 0.00 | 0.67 / 0.50 |
| 4 | gpt-3.5-turbo | ✓ | | ✓ | | 0.68 / 0.60 | 0.46 / 0.40 | 0.23 / 0.17 | 0.17 / 0.17 | 0.33 / 0.50 |
| 5 | gpt-3.5-turbo | ✓ | | | ✓ | 0.57 / 0.47 | 0.46 / 0.30 | 0.00 / 0.32 | 0.17 / 0.17 | 0.17 / 0.17 |
| 6 | gpt-3.5-turbo | ✓ | | ✓ | ✓ | 0.58 / 0.65 | 0.34 / 0.47 | 0.23 / 0.13 | 0.17 / 0.17 | 0.33 / 0.33 |
| 7 | gpt-3.5-turbo | ✓ | ✓ | ✓ | ✓ | 0.52 / 0.59 | 0.38 / 0.41 | 0.19 / 0.15 | 0.17 / 0.17 | 1.00 / 0.67 |
| 8 | Llama-2 | | | ✓ | | 0.68 / 0.61 | 0.39 / 0.27 | 0.00 / 0.00 | 0.17 / 0.00 | 0.17 / 0.17 |
| 9 | Llama-2 | | | | ✓ | 0.41 / 0.62 | 0.00 / 0.34 | 0.00 / 0.00 | 0.00 / 0.00 | 0.00 / 0.17 |
| 10 | Llama-2 | | | ✓ | ✓ | 0.61 / 0.55 | 0.33 / 0.22 | 0.00 / 0.00 | 0.17 / 0.00 | 0.17 / 0.33 |
| 11 | Llama-2 | | ✓ | ✓ | ✓ | 0.65 / 0.63 | 0.35 / 0.26 | 0.23 / 0.27 | 0.33 / 0.00 | 0.33 / 0.00 |
| 12 | Random | | | | | 0.50 | 0.34 | 0.17 | 0.17 | 0.60 |
| 13 | Human | ✓ | ✓ | ✓ | ✓ | 0.76 | 0.72 | 0.33 | 0.5 | 0.67 |

tuned versions of GPT-3.5 and Llama-2. We find that fine-tuned GPT-3.5 performs well (micro-f1: 0.43) when predicting the strategies for each label, followed by the vanilla version of GPT-4 (micro-f1: 0.37). However, Llama-2-13b underperforms (micro-f1: 0.15), even in the fine-tuned version (micro-f1: 0.20).

## 5.2 Turn Order and Utterances

For the 20 games we conducted, roles have been randomly assigned to each of the six players so as not to bias the data towards certain advantageous positions of Merlin or evil players in the turn order. We found that identifying good players is easier if Merlin is among the first three players (p-value 0.039). Similarly, if Morgana or Percival speaks a lot, good players are easier to identify (p-value 0.046 and 0.021, respectively). We also found that evil players who uttered lies more frequently had a statistically significant advantage of identifying Merlin correctly (p-value 0.015). We conjecture that Merlin's attempts to counteract the lies of evil players has a high likelihood of revealing his identity to evil players. Similarly, we found a statistically significant influence of the number of Percival's utterances and the likelihood of evil winning (p-value 0.018).

## 5.3 Comparing Model Performance

We compare three state-of-the-art LLMs, namely GPT-4 (OpenAI, 2023), GPT-3.5-turbo, and Llama-2-13B (Touvron et al., 2023), as well as a fine-tuned version of Llama-2-13B and GPT-3.5-turbo. Each model is used to predict the roles of all players from the perspective of an external observer, but also on the task of identifying Merlin, given the privileged knowledge $e$ of who the evil players are. Each evaluation is conducted with three different sets of modalities, including the game's state, player chat, and a combination of game state and player chat. Our inherent assumption is that these LLMs possess enough encoded knowledge in their pre-trained model weights such that inference for our task is possible. To assess the pre-trained model's capability, we evaluate each model's familiarity with the rules of *Avalon*. We run three prompts and list selected answers in Table 5. Responses to all three prompts are evaluated by a human for correctness and adherence to the game's rules and are listed in Table 2.

Table 2 demonstrates the performance of various LLM approaches. Due to the probabilistic nature of the LLMs, each model was evaluated ten times on each of the validation games. Human performance (line 13) was evaluated by asking three human annotators for each of the six evaluation games to provide their estimates about the roles of each player in each game. Further, the performance of finding Merlin and identifying Merlin correctly at any point in the game was directly taken from the game's belief annotations, while the final prediction was taken either from the assassin's choice (if applicable) or the last anytime estimate. Table 2 presents the F1-score when providing information about the game in a round-based context with a carried-over belief (first number) and when providing the entire context from the beginning of the game (second number). In comparison, the round-

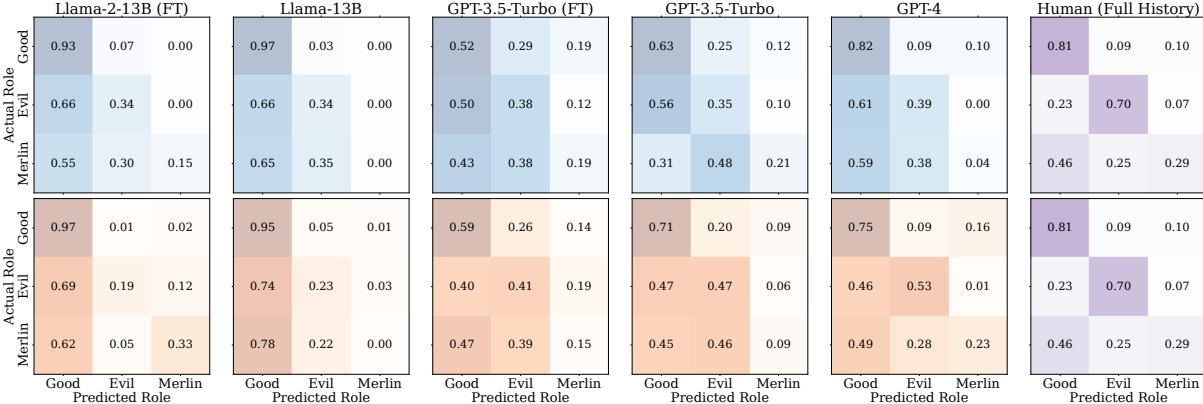

Figure 4: Confusion matrices evaluating our LLMs in the two context conditions using Chat+State: round-based (top, blue) and full-context (bottom, orange). Human results (right, purple) are for the full context in both cases.

based context utilizes an average of 974 (std: 333, max: 1941) tokens, while the full context utilizes an average of 2844 (std: 2011, max: 8556) tokens.

While most models and modalities outperform randomly guessing player roles, only GPT-4 in the round-based context utilizing chat (line 1) outperforms the human baseline (line 13) when identifying Merlin among the players. Comparing each model's performance when identifying good and evil players, we notice a performance drop for evil players as they actively hide their identity to appear as good players. A similar trend can be observed in the human baseline (line 13), however, humans are unmatched in their ability to identify them. Generally, this trend is also observed, to a greater extent, in the ability to identify Merlin.

For GPT-based models, when predicting good and evil roles, we notice that combining game chat and game state information results in the highest F1 scores (lines 3 and 6) when providing the full context, confirming our hypothesis that grounding game chat in the game's state improves performance. However, when utilizing the round-based context, individual modalities tend to outperform their full-context equivalents (lines 1, 2, 4, and 5), yet do not reach the performance of the joint modality. Further, we observe that predicting Merlin benefits from the availability of the game state (lines 2 and 5).

In the case of Llama-2, the best performance is achieved when only utilizing individual modalities (lines 8 and 9) while not allowing any conclusive insight as to which modality tends to perform better, despite significant differences (line 9 (good) and line 10 (evil)) in their respective performance).

Figure 4 provides further insights into the role

predictions, particularly the two different context modes (round-based and full-context) when using the joint modality. Particularly with round-base context, GPT-3.5-Turbo (Figure 4, top) suffers by mostly confusing evil players with good ones and identifying Merlin as evil. On the other hand, providing the full context (Figure 4, bottom), does not lead to such confusion about Merlin, where he is mostly confused with a good player. We conjecture that identifying players' underlying intentions, especially when players actively try to hide their identity, remains a challenging task for LLMs.

### 5.3.1 LLM Fine-Tuning for Role Prediction

In addition to testing the performance of pre-trained models, we have fine-tuned *GPT-3.5-Turbo* as well as *Llama-2-13b*. Training data for fine-tuning was generated from the remaining 14 games of Avalon that were not used for evaluation. We have trained each model for three epochs across these 14 games and report their performance in Table 2 and Figure 4. Particularly in the case of Llama-2, we notice that fine-tuning improves the model's performance when predicting the roles of all players. For GPT-3.5, we do not observe the same trend; however, when predicting Merlin on its own, GPT-3.5 dramatically improves in performance, even outperforming GPT-4 and the human baseline. When comparing the failure cases in Figure 4, fine-tuning reduces failure severity, particularly for Merlin, limiting failed predictions to good players, particularly in the case of Llama-2.

### 5.4 Comparison to other Deception Tasks

We have also compared the prediction capabilities of pre-trained models on existing datasets of social-deduction games, namely the game of Were-

Table 3: F1 scores for role predictions in the game of Werewolf. Our results show enhanced role prediction scores compared to our *Avalon* data, suggesting a more challenging social deduction environment in *Avalon*.

| | Model | Tuned | Good | Evil | Seer | Gain |
|---|---|---|---|---|---|---|
| | | | | Roles | | |
| 1 | GPT-4 | | 0.66 | 0.66 | 0.49 | +0.28 |
| 2 | GPT-3.5-Turbo | | 0.53 | 0.56 | 0.35 | +0.19 |
| 3 | GPT-3.5-Turbo | ✓ | 0.61 | 0.72 | 0.43 | +0.61 |
| 4 | Llama-2-13B | | 0.43 | 0.47 | 0.00 | +0.13 |
| 5 | Llama-2-13B | ✓ | 0.45 | 0.39 | 0.08 | -0.24 |

wolf (Lai et al., 2022). Table 3 shows the results in a similar setting, grouping the players into Good, Evil, and Seer by designating roles as Evil if lying and deception are a part of their strategy and keeping the Seer as a separate role akin to Merlin in *Avalon*. Predictions for individual roles with the best model, GPT-4, are available in Table 7.

We demonstrate that in the setting of Werewolf, LLMs perform well predicting the players' roles (see "gains" in Table 3), which we hypothesize is due to the shorter horizon (Avg. 1786 tokens, 2795 max) of each game as compared to our *Avalon* dataset (Avg. 2844 tokens, 8556 max), making the task of identifying roles easier. We find that fine-tuning dramatically improves the performance of GPT-3.5 over three epochs (line 3 vs. line 2) of training, while Llama-2-13B fine-tuning does not significantly affect performance (line 5 vs. line 4). This observation is contrary to our Avalon dataset (see Table 2 lines 7 and 11), where Llama-2, in particular, improved its performance, while GPT-3.5 roughly maintained its performance, resulting in a large difference in gains (+0.61 vs. -0.24). We hypothesize that with the high-quality data provided in our Avalon dataset, fine-tuning Llama-2 allows it to achieve performance levels comparable to an untuned GPT-3.5 (see Table 2). Yet, when faced with the Werewolf dataset, a 'naturalistic-social-setting' characterized by frequent cross-talk and off-topic exchanges, Llama-2 struggles to fine-tune effectively. In contrast, the broad training foundation of GPT-3.5 enables it to learn even under such conditions.

## 6  Conclusion

In this work, we present a novel benchmark, associated dataset, and testbed for long-horizon dialogue understanding in scenarios of conflicting interests among multiple participants. This task combines utterances from six human players at a time, hand-labeled persuasion and deception strategies, player beliefs, and comprehensive game states recorded over more than 24 hours of gameplay. We demonstrate that current state-of-the-art LLMs do not reach human-level performance in environments that require the understanding and tracking of *long-horizon* dialogue between multiple participants in challenging social cooperative-competitive settings. Compared to similar datasets, our benchmark contains longer context horizons, stricter game rules, and high-quality dialogue, making it well-suited for NLU research as all game-relevant communication has been captured and is thus, available to learning algorithms. This dataset opens doors for diverse research avenues, from detecting deception to developing conversational agents for cooperative-competitive settings. We hope that our high-quality dataset will be valuable to further tune and evaluate the capabilities of large language models in challenging social settings.

## Limitations

The performance of our evaluated models depends on the capabilities of the pre-trained language models. Given our cooperative-competitive multi-party scenario, we observe that current approaches do not reach human-level performance. We also believe that the fine-tuning results could be further improved by extending our dataset; however, we believe that future work will utilize it successfully.

## Ethics Statement

Participants in our IRB approved human-subject study were provided consent forms and privacy notices explaining how we intend to use the data collected during the experiments. We strictly adhere to applicable data protection policies and use the data solely for research purposes. Further, we offered opportunities for participants to seek clarification and ask questions, fostering informed consent and ethical, trustworthy interactions.

## Acknowledgements

This work has been funded in part by the Air Force Office of Scientific Research (AFOSR) under grants FA9550-18-1-0251 and FA9550-18-1-0097, the Army Research Laboratory (ARL) under grant W911NF-19-2-0146, DARPA award HR001120C0036, and the Office of Naval Research (ONR) award N00014-23-1-2840.

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

## A Player Instructions

The following sections provide further detail about the rules of Avalon, as well as the hidden knowledge imbued to every special role in the game.

### A.1 In-Game Instructions

#### A.1.1 How to Play

*Avalon: The Resistance* is the game of hidden loyalty. Players are either Loyal Servants of Arthur fighting for Goodness and honor or aligned with the Evil ways of Mordred. Good wins the game by successfully completing three Quests. Evil wins if three Quests end in failure. Evil can also win by assassinating Merlin at game's end or if a Quest cannot be undertaken. Players may make any claims during their turns. Discussion, deception, accusation, and logical deduction are all equally important in order for Good to prevail or Evil to rule the day. Before playing a game, please check out the official rules. The differences in this version of *Avalon* in comparison to these rules will be explained below.

#### A.1.2 Roles

In this version of *Avalon*, you will play with five fixed roles:

**Merlin** Merlin is on Good's side and knows the two evils' identity. The portrait frames of the evil people will have a red circle around them.

**Percival** Percival is on the side of Good and gets information about two players. These two players are Morgana and Merlin; however, Percival does not know who is who. These two players will have a red-and-blue circle around them.

**Servants** Servants are on the side of Good but do not have any special knowledge or ability.

**Morgana** Morgana plays on the side of Evil and knows who the other evil player is, indicated by a red circle around their profile picture. Morgana appears as a potential Merlin to Percival.

**Assassin** Assassin plays on the side of Evil and knows who the other evil player is, indicated by a red circle around their profile picture. At the end of the game, should Good win, the Assassin can win the game for Evil by correctly identifying who Merlin is.

#### A.1.3 Communication

For the purposes of this research study, communication between the players will be conducted through turns. During each player's turn, they will be able to communicate through the chat window, responding to previous questions or accusations, making new statements, and asking questions to other players. However, only the player whose turn it is will be able to use the chat. Please only communicate with the players within the interface provided in this version of the game.

After you send a message to the chat, you can indicate the strategy/motivation you used when writing the message. While not mandatory, it would be great if players would provide these insights. The information provided in the strategy selection will not be shared with other players, so don't worry about saying that you potentially lied about something!

#### A.1.4 Communication Labels

These are the labels and explanations available for your communication

**Assertion** his includes statements where the speaker makes an assertive remark, expresses a firm belief or makes a definitive statement.

**Questioning** This includes any questioning of other players regarding their behavior.

**Suggestion** This includes all instances where the speaker suggests an action, makes a proposal, or gives advice to another player.

**Agreement** This encompasses sentences where the speaker is agreeing with another player's statement or strategy, affirming their own identity or someone else's.

**Logical Deduction** This includes statements where the speaker provides a reasoned explanation, defends a point of view, elaborates a strategy or justifies their actions.

**Compromise/Concession** This category includes any sentences where the speaker concedes to another's point of view, expresses indecision, or appears to back down from a previously held stance.

**Critique/Opposition** This includes instances where the speaker critiques another's view, counters an argument, or points out an inconsistency or flaw in another's reasoning.

**Appeal/Defense** This category covers situations where the speaker appeals to others for trust, to be

included in the quest, or uses emotional/personal appeal to gain favor.

### A.1.5 Turns

The player with the crown is the current quest leader, while the jester's hat indicates whose turn it currently is. Each turn is limited to 100 seconds, after which the game automatically transitions to the next player. Similarly, if a vote is necessary, you will have 30 seconds to cast your vote. After that, parties/quests will be approved automatically for players that didn't vote.

The quest leader has to allow one round of discussion prior to being able to initiate a party vote. However, party proposals can be changed whenever it is the turn of the leader.

### A.1.6 Parties

Parties are indicated by a little shield icon next to a player's profile. If there is a shield, players are part of the party.

### A.1.7 Selecting Players

Players can be selected for a party or by the Assassin to indicate who they think is Merlin by clicking on their player profile picture frame. However, remember to confirm your choices. Clicking on player profile frames alone will only count as a choice if the choice is explicitly confirmed.

### A.2 Differences to Standard Avalon

To run the games effectively in an online fashion, we incorporate a few additional rules that are outlined in this section. Particularly, discussions are conducted in a turn-based manner, enforcing that only one player can talk at a time. Conversations are captured through a chat-based interface, ensuring that all game-related player interactions are captured precisely. A player's turn ends when either they choose to end their turn via an "End my Turn" button or when their turn-time of 200 seconds runs out. In either case, the next player will start their turn. However, if the turn time runs out for the current quest leader, instead of transitioning to the next player, a random party is proposed, or a vote for the currently proposed party is initiated, depending on the current game state. In addition to ending their turn, quest leaders have the choice to initiate another round of discussion or to initiate a vote for the currently proposed party. Similar to the turn-time, votes (either for a party or for quest success) have to be made within 30 seconds. Should

the time run out prior to a decision, parties, and quests will automatically be approved or succeeded. Lastly, if the outcome of the game requires it, the assassin has 200 seconds to evaluate the game's chat history and decide on who they think is Merlin, and conduct an assassination. Failure to choose a player as potential Merlin will automatically fail the vote, proceeding to a victory for the good players.

### A.3 Human Baseline Labeling Instructions

On the following page, you will get see the log of a game of *Avalon*, including a description of the current game state in the form of the "system" messages. Please provide your role estimates for each player.

- After each round of discussion, you will have the opportunity to indicate your beliefs about the role of each player.

- Take your time to carefully analyze the available information and make your best judgments about each player's role.

- Avoid accessing or considering information from future rounds or pages when making your decisions. Focus only on the information available up to the current round.

- Once you have made a belief label for a player in a particular round, refrain from changing that label later, even if you change your mind about it in subsequent rounds.

## B Surveys

The following section introduces the surveys conducted with our human subjects to collect the baseline results.

### B.1 Example of Human Baseline Survey

This is an example of the human baseline questionair

### B.1.1 Example for Round 1
**Round 1**

- **system: Game Started!**

- **system: player-2 proposed a party: player-2, player-3**

- player-2: This is the first round, we don't have a lot of information so I

- propose myself (because I am good) and the next person player-3.

- player-3: I agree with player-2. I am good too.

- player-4: I think I am fine with the team as well

- player-5: i am good with that

- player-6: No opinions in the first turn. I'm fine with this party

- player-1: Oh, yeah, I am good with this. We don't know anything at this point anyways

- **system: Party Vote Outcome: player-1: Yes, player-2: Yes, player-3: Yes, player-4: Yes, player-5: Yes, player-6: Yes**

- **system: Vote Succeeded! Initiating Quest Vote!**

- **system: Quest Succeeded!**

*Quest results after quest 1: Success*

———————

Which role do you think the players have? Please only consider the information you have up to this point in the game.

- player-1: <select one of ["good", "evil", "merlin", "I don't know"]

- player-2: <select one of ["good", "evil", "merlin", "I don't know"]

- player-3: <select one of ["good", "evil", "merlin", "I don't know"]

- player-4: <select one of ["good", "evil", "merlin", "I don't know"]

- player-5: <select one of ["good", "evil", "merlin", "I don't know"]

- player-6: <select one of ["good", "evil", "merlin", "I don't know"]

## C  Further Prompts

In this section, we provide additional insights into how we query the LLM.

### C.1  Identifying Merlin

The following prompt was utilized to query the model about the identity of Merlin

**system:** You are a helpful assistant that uses the chat between six players, player-1 to player-6, who play Avalon: The Resistance (a cooperative-competitive game) to identify who is Merlin. There are three good players, Merlin, who is also a good player, and two evil players. Evil can usually be found because they are deceptive and lie about the good player's roles and vote for parties and quests irrationally. For Merlin, watch out for individuals with knowledge of evil players' identities, insightful comments beyond their role, and caution regarding mission teams or specific players.
**human:** The current state is: *<STATE>* (See Section 4.2.1)
The current party proposal is: *<PARTY>* (See Section 4.2.1)
Your initial belief is: *<BELIEF>* (See Section 4.1, only valid in round-based context)
You know that <EVIL> are evil.
This is the chat between player-1 to player-6:
*<CHAT>* (See Section 4.2.2)
Who do you think is Merlin? Please do not explain your answer, do not elaborate on it further.
**assistant:**

Table 4: Prompt used to infer who Merlin is.

**human:** Can you explain the rules of Avalon: The Resistance to me?
**Assistant (gpt-3.5-turbo):** Avalon: The Resistance is a board game that involves hidden roles, deduction, and deception. The game is set in the world of King Arthur, and players take on the roles of either loyal servants of Arthur (the "good" faction) or the minions of the evil Mordred (the "bad" faction). The goal of the game is different for each faction. [...]
**human:** In Avalon: The Resistance, can you tell me about the role of Merlin?
**assistant (Llama-2-13B):** [...] Merlin is a character with special knowledge who knows the true identities of the minions of Mordred, the traitors in the game. Here are the key points to understand about the role of Merlin: [...]
**human:** In Avalon: The Resistance, what is a good strategy for Morgana when playing with Percival and Merlin?
**assistant (gpt-4):** A good strategy for Morgana when playing with Percival and Merlin in Avalon: The Resistance is to focus on confusing and deceiving both Percival and Merlin to turn the tide in favor of the evil team. Here are some tips to execute this strategy: [...]

Table 5: Prompts and sample answers to determine a model's familiarity with *Avalon: The Resistance*.

### C.2  Familiarity with Avalon

The questions in Table 5 are used to identify whether or not a particular pre-trained LLM is aware of *Avalon: The Resistance*. Sample answers indicate that all three tested models (GPT-4, GPT-3.5, Llama-2-13B) are familiar with Avalon.

### C.3  TypeScript Schemas

The following schema is used for role predictions:

```
// Define the AvalonRoles interface
export interface AvalonRoles {
    player_1: "good" | "evil" | "merlin";
    player_2: "good" | "evil" | "merlin";
    player_3: "good" | "evil" | "merlin";
    player_4: "good" | "evil" | "merlin";
    player_5: "good" | "evil" | "merlin";
    player_6: "good" | "evil" | "merlin";
}
```

The following schema is utilized to predict the player who is playing as Merlin:

```
// Define the MerlinPlayer interface
export interface MerlinPlayer {
```

Table 6: F1 scores for role predictions in the game of Werewolf. Our results show enhanced role prediction scores compared to our *Avalon* data, suggesting a more challenging social deduction environment in *Avalon*.

| | Strategy | Model | | | | |
|---|---|---|---|---|---|---|
| | | GPT-4 | GPT-3.5 | GPT-3.5 (FT) | Llama-2-13b | Llama-2-13b |
| 1 | Assertion | 0.27 | 0.26 | 0.26 | 0.25 | 0.26 |
| 2 | Questioning | 0.40 | 0.33 | 0.39 | 0.16 | 0.00 |
| 3 | Suggestion | 0.57 | 0.11 | 0.56 | 0.12 | 0.00 |
| 4 | Agreement | 0.54 | 0.40 | 0.55 | 0.24 | 0.00 |
| 5 | Logical Deduction | 0.36 | 0.41 | 0.42 | 0.00 | 0.26 |
| 6 | Compromise/Concession | 0.17 | 0.40 | 0.23 | 0.18 | 0.00 |
| 7 | Critique/Opposition | 0.36 | 0.43 | 0.43 | 0.07 | 0.33 |
| 8 | Appeal/Defense | 0.12 | 0.00 | 0.42 | 0.00 | 0.00 |
| 9 | Overall (micro-f1): | 0.37 | 0.32 | 0.43 | 0.15 | 0.20 |

Table 7: F1 scores for Werewolf role predictions across test splits using the optimal model (GPT-4) from Table 3.

| Model | Drunk | Insomniac | Minion | Hunter | Revealer | Villager | Werewolf | Troublemaker | Robber | Tanner | Seer | Mason |
|---|---|---|---|---|---|---|---|---|---|---|---|---|
| 1 GPT-4 | 0.42 | 0.70 | 0.04 | 0.43 | 1.00 | 0.85 | 0.41 | 0.71 | 0.39 | 0.30 | 0.55 | 0.00 |

```
    merlin: "player_1" | "player_2" |
       "player_3" | "player_4" |
       "player_5" | "player_6";
}
```

# D  Additional Results

## D.1  Werewolf Individual Roles

In Table 7, we demonstrate that many key roles in Werewolf are predicted with high F1 scores. However, Mason and Minion have low scores as they only have one and four representations in the test set of [2]. For Tanner and Robber, we observe a common confusion with the Werewolf role (and vice versa). We attribute this to the Tanner's goal of looking like a werewolf and being eliminated for their unique win condition, while the Robber (at the end of the game) may have played like a werewolf without being aware of the fraction change.

## D.2  Persuasion Strategy Prediction

Table 6 reports the performance of our model when predicting the persuasion strategies for all utterances across our six test games. In each case, models have been run once for each utterance, while the fine-tuned models have been trained for three epochs on the 14 training games. We observe that a fine-tuned GPT-3.5 model performs best, even outperforming a vanilla GPT-4 model, while Llama-2-13B, even in the fine-tuned case, underperforms.