# OpenReview forum: "Long-Horizon Dialogue Understanding for Role Identification in the Game of Avalon with Large Language Models"
_EMNLP/2023/Conference — EMNLP 2023 Findings_

### Official Review · Reviewer_f9MG · 2023-07-31

**Soundness:** 3

**Excitement:**

3: Ambivalent: It has merits (e.g., it reports state-of-the-art results, the idea is nice), but there are key weaknesses (e.g., it describes incremental work), and it can significantly benefit from another round of revision. However, I won't object to accepting it if my co-reviewers champion it.

**Missing References:**

N/A

**Paper Topic And Main Contributions:**

This paper introduces a dataset simulating *Avalon: The Resistance* by developing a browser version of the game. This simulation includes multi-party conversation (i.e., 6 participants in each game) with global information. Through this dataset, the authors try to explore LLMs' ability to understand long-horizon dialogue in complex situation where people have different roles and objectives.

**Questions For The Authors:**

- Question A: Do the authors have information about how often players speak in a single round? Would there be any potential bias in terms of the speaking order?

**Reasons To Accept:**

Exploring language understanding ability of language models through existing games is a nice motivation in that games simulate human interaction well. One relevant example not mentioned in this paper would be [Dungeons and Dragons dataset](https://aclanthology.org/2022.emnlp-main.637/) for dialogue challenge. The Avalon game also contains many attractive factors for understanding human interaction.

### After rebuttal
I appreciate the authors for handling all of my concerns in detail. Based on the score rubric, I would raise my Soundness score to 3.

**Reasons To Reject:**

I have concerns about this paper as following:

**[1] Lack of analysis**

My major concern is that this paper skips the analysis of model performance, which is one of the most important part. Most experimental results describe the numerical performance of subtasks among models, according to input type. Although one of the main motivation of the authors is "the presence of bad faith actors and deception (as mentioned in Introduction)", there is no analysis of how these kinds of bad faith actors and deception actually affect the model performance. It has to be analyzed because there would also be several factors affecting the poor performance of language models. I recommend to clearly clarify the research question of the paper.

**[2] Composing model input using the proposed dataset**

Although the authors try to handle multi-turn conversation, the input given to the model only contains utterances of the last turn and previous information is only expressed as structured information. To the best of my knowledge, this can cause two issues. First, not containing previous utterances cannot sufficiently capture the interactive features of human conversation because natural language context implicitly draw out extra information, such as nuance and mind state. Second, replacing previous information with structured state can be suboptimal. In the perspective of dialogue state tracking, [this paper](https://aclanthology.org/2021.emnlp-main.593/) shows that using belief state rather than actual conversational history can deteriorate performance. Therefore, it is hard to fully understand the performance of multi-turn modeling.

**[3] Scalability of the dataset construction framework**

The authors construct a web version of the game and annotated the dataset manually. This is valuable platform to the subfield, but it also has a limitation because annotating additional dataset consistently would be costly. But I think this is not a major concern compared to concerns above.

**Reproducibility:**

4: Could mostly reproduce the results, but there may be some variation because of sample variance or minor variations in their interpretation of the protocol or method.

**Reviewer Confidence:**

4: Quite sure. I tried to check the important points carefully. It's unlikely, though conceivable, that I missed something that should affect my ratings.

**Typos Grammar Style And Presentation Improvements:**

- Captions of Table 2 and Table 5 need to be explained more concretely because the content is the same even though they indicate difference situations.

---

> ### Author Rebuttal · Authors · 2023-08-29
>
> We thank the reviewer for their thoughtful comments and are thrilled to hear that our testbed has been deemed valuable for the subfield and that Avalon has many attractive factors to investigate human interaction. We also thank the reviewer for pointing us to [1], and we have added the citation to our paper.
>
> **Highlights:**
> - We have added deception strategies (omission, commission, influence [2]) for each evil utterance that is identified as a lie.
> - We have added additional analysis with regards to the influence of player order and utilized lies on the model performance and outcome of the game
> - We added experiments utilizing the full game context with all models for better comparison of game representations and report our results as F1 scores (see large table below)
> - We have removed Vicuna in favor of Llama-2 and added GPT-3.5 and Llama-2 finetuning
> - We have added experiments across all models to predict persuasion strategies (see small table below)
>
> **Lack of analysis**: We have conducted additional analysis with regard to player ordering in the game as well as deception strategies (i.e., lies) utilized by the evil players. In particular, we appreciate the request for insights about bad-faith actors. In order to address this, we have added additional deception strategy labels to our dataset for each utterance of the evil players (711 utterances total) with respect to being neutral, or a type of lie (commission, omission, or influence [2]), out of which 429 utterances constitute a lie. For our analysis, we have investigated the model’s performance and game outcome on: the number of utterances per player type, the number and type of lies, and the position of Evil players and Merlin in the turn order. With these types of evaluations, we hope to address the question of how bad-faith actors influence the gameplay. We have found that Merlin is easier to identify when Evil has a high number of utterances being labeled as lies (particularly Commission), while it is more likely that Good will lose if Percival contributes a large number of utterances. Similarly, good players are easier to identify by our model if Percival or Morgana contribute many utterances, or if Merlin is early in the turn order. We have added further analysis to this end to our paper and supplemental material.
>
> **Composing model input using the proposed dataset**: Since our initial paper submission, additional LLMs have been released that allow for extended contexts during prompting (Llama-2), as well as additional models that can now be fine-tuned (GPT-3.5-turbo) exhibiting large contexts. We have followed the reviewer’s suggestion and added experiments that utilize the full game context (without a carried structured belief) for all models across all experiments and reported results in our new and updated Table 3 (see large table below). In comparison, the round-based context utilizes an average of 974 (std: 333, max: 1941) tokens, while the full context utilizes an average of 2844 (std: 2011, max: 8556) tokens. For this, we find that the full context does not necessarily result in a blanket improvement of performance for any model. However, there is no clear picture that would allow for a final conclusion. With this, we would like to highlight again the purpose of our contribution. In particular, we introduce a new, complementary benchmark for long-horizon deceptive tasks with hidden agendas in a multi-party cooperative-competitive game that poses significant challenges to current SOTA LLMs. As humans are influenced by and utilize deception, we hope to foster further research into how such behavior can be identified and understood.
>
> **Scalability of the dataset** construction framework: With respect to scalability, the provided testbed is fully automated and does not require the intervention of an administrator. During our gameplay, players self-labeled 82% of their utterances with persuasion strategies outside of their own turn, thus also keeping players engaged in the game while they wait for other players’ turns to end. We hand-labeled the remaining 18% of utterances after the fact (2348 total labels). Additionally, the newly added deception labels were all hand-labeled by a small group of annotators that had prior familiarity with the data. We believe that the provided online version of Avalon is valuable as it can operate in an unattended fashion. Data is collected and processed automatically, and persuasion strategies are largely self-labeled by players, making it easy to further extend the dataset with new games, or different team compositions.
>
> In the following, we would like to answer your questions:
>
> **How often do players speak**: Across our dataset of 20 games, the six players went on 102 quests, making 768 turns (one quest can have multiple rounds of discussions). Out of these 768 turns, 9 players chose not to speak, skipping their turn without providing an utterance. From observation, these skipped turns happen when multiple rounds of discussions are happening for the same quest.
>
> **Bias in the speaking order**: Thank you for asking this interesting question! We have analyzed the speaker order with regard to the game outcome and model performance; however, we have not found a statistically significant impact on the win rate or model performance related to player order (we evaluated both evil players being in the first/last three players, evil players being next to each other, Merlin being in the first/last three players, or merlin being surrounded by evil players). However, given Percival's behavior, we found an impact on game outcome and model performance. Good players tend to lose the game if many utterances originate from Percival. Furthermore, if Percival or Morgana are getting very involved (i.e., producing many utterances) or if Merlin is early in the turn order, it is easier for the model to identify good players. Finally, we identified that if evil players utilize many commission utterances, the model’s ability to identify Merlin increases dramatically.
>
> **Additional Results:**
>
> **Prediction of Persuasion Strategies:**
> | PStrat | GPT-4 | GPT-3.5 |  GPT-3.5 (FT) | Llama-2-13b | Llama-2-13b (FT) |
> |----------|-------|---------|---------------|-------------|------------------|
> | Micro-F1 | 0.37  | 0.35    | 0.43          | 0.15        | 0.20             |
>
> **Updated Table 3 (Role Prediction)**
> In the table below, the first number represents the round-based inference with a carried-over belief, while the second number represents inference with the full context.
>
> | Model | Familiar with Avalon | Trained Model (14 Games) | Using Chat | Using State | Find Good | Find Evil | Find Merlin | Evil find Merlin (Final Guess) | Evil find Merlin (Anytime)  |
> | -- | -- | -- | -- | -- | -- | -- | -- | -- | -- |
> | Gpt-4      | [x] | | [x] | | 0.67 / 0.67 | 0.48 / 0.55 | 0.36 / 0.20 | 0.17 / 0.17  | 0.83 / 0.67  |
> | Gpt-4      | [x] | | | [x] | 0.59 / 0.57 | 0.20 / 0.33 | 0.06 / 0.31 | 0.17 / 0.33 | 0.17 / 0.50 |
> | Gpt-4      | [x] | | [x] | [x] | 0.67 / 0.68 | 0.46 / 0.58 | 0.05 / 0.27 | 0.00 / 0.00 | 0.67 / 0.50 |
> | gpt-3.5-turbo      | [x] | | [x] | |  0.68 / 0.60 | 0.46 / 0.40 | 0.23 / 0.17 | 0.17 / 0.17 | 0.33 / 0.50 |
> | gpt-3.5-turbo      | [x] | |  | [x] | 0.57 / 0.47 | 0.46 / 0.30 | 0.00 / 0.32 | 0.17 / 0.17 | 0.17 / 0.17 |
> | gpt-3.5-turbo      | [x] | | [x] | [x] | 0.58 / 0.65 | 0.34 / 0.47 | 0.23 / 0.13 | 0.17 / 0.17 | 0.33 / 0.33 |
> | gpt-3.5-turbo      | [x] | [x] | [x] | [x] | 0.52 / 0.59 | 0.38 / 0.41 | 0.19 / 0.15 | 0.17 / 0.17 | 1.00 / 0.67 |
> | Llama-2 |  | | [x] | | 0.68 / 0.61 | 0.39 / 0.27 | 0.00 / 0.00 | 0.17 / 0.00 | 0.17 / 0.17 |
> | Llama-2 |  | | | [x] | 0.41 / 0.62 | 0.00 / 0.34 | 0.00 / 0.00 | 0.00 / 0.00 | 0.00 / 0.17 |
> | Llama-2 |  | | [x] | [x] | 0.61 / 0.55 | 0.33 / 0.22 | 0.00 / 0.00 | 0.17 / 0.00 | 0.17 / 0.33 |
> | Llama-2 | [ ] | [x] | [x] | [x] | 0.65 / 0.63 | 0.35 / 0.26 | 0.23 / 0.27 | 0.33 / 0.00 | 0.33 / 0.00 |
> | Random |  |  |  |  | 0.50 | 0.34 | 0.17  | 0.17 | 0.60 |
> | Human | [x] | | [x] | [x] | 0.76 | 0.72 | 0.33 | 0.5 | 0.67  |
>
> 1. Callison-Burch et al.: “Dungeons and Dragons as a Dialog Challenge for Artificial Intelligence”
> 2. The Three Types of Lies: Penn State University

---

### Official Review · Reviewer_n5L7 · 2023-08-03

**Soundness:** 4

**Excitement:**

3: Ambivalent: It has merits (e.g., it reports state-of-the-art results, the idea is nice), but there are key weaknesses (e.g., it describes incremental work), and it can significantly benefit from another round of revision. However, I won't object to accepting it if my co-reviewers champion it.

**Paper Topic And Main Contributions:**

This paper introduces a new dataset for dialogue understanding in the hidden role game *Avalon: The Resistance*.
In a hidden role game, players are assigned to teams with competing goals (accomplished via voting) which is dependent upon their hidden role.
They are forbidden from communicating their role aloud, so team members may not know they are on the same team.
These games often require persuasion and deception on the part of the players, which makes identifying roles in the dialogue a challenging task.
Previous datasets from hidden role games exist for the game *Mafia* [1] (a dataset containing dialogue and voting history) and for *One Night Werewolf* [2] (a multimodal dataset with video and annotation of the persuasion strategy, in addition to dialogue and voting history).
The introduced dataset for *Avalon* includes dialogue from the game chat and self-reported persuasion strategies (rather than strategies annotated by third parties as in [2]).

Given this dataset, the paper describes two tasks specific to *Avalon*: identifying all roles (good, evil, or Merlin) and identifying the role of Merlin at each stage of the game given knowledge of the evil characters.
A prompting strategy for encoding player chat and game state are devised.
Three LLMs are experimented with for solving the tasks: GPT-3.5/4 and Vicuna (both pre-trained and fine-tuned).
Experiments also ablate both the chat and game state, providing three combinations of experiments for each LLM.
These are compared against both a human and random baseline.
Interestingly, both pre-trained Vicuna and GPT-3.5 outperform GPT-4 at identifying all roles when ablating game state.
On the other hand, GPT-4 performs nearly as well as humans for identifying Merlin at the end of the game and outperforms humans at identifying Merlin at any stage of the game.
Several models perform worse when provided both game state and chat and the fine-tuned Vicuna severely underperforms the pre-trained version in all settings.

### Update after Rebuttal

I've raised my excitement score from 2 to 3, mostly because the authors mention that the previously released Avalon test set from [2] is not currently available due to intellectual property issues. I've also raised my soundness to 4 as the authors have:

1. Clarified the issues I brought up in my initial review regarding things like use of average precision rather than F1.
2. Addressed how their dataset is more complex than previous datasets through additional experimental results comparing against Werewolf dataset [2] (these additional results should be included in an updated version of their paper).

[1] Samee Ibraheem, Gaoyue Zhou, and John DeNero. NAACL 2022. Putting the con in context: Identifying deceptive actors in the game of mafia.
[2] Bolin Lai, Hongxin Zhang, Miao Liu, Aryan Pariani, Fiona Ryan, Wenqi Jia, Shirley Anugrah Hayati, James M Rehg, and Diyi Yang. Findings of ACL 2023. Werewolf among us: A multimodal dataset for modeling persuasion behaviors in social deduction games.

**Questions For The Authors:**

* Can an utterance have multiple labels? If not, why not? It seems like a single utterance can employ multiple persuasion strategies.
* Why did you need to use a spell checker? What about the use of slang, emoticons, etc?

**Reasons To Accept:**

This paper introduces a new dataset for the hidden role game *Avalon: The Resistance* which includes self-reported play strategies from the actual players (rather than having third-parties annotate strategies).

**Reasons To Reject:**

There are numerous weaknesses in this paper:

* A major issue is one of novelty. I'm usually quite loathe to bring up novelty as an issue, but in this case I believe it is warranted. There are already two dialogue datasets, [1] and [2], which explore dialogue understanding in hidden role games. [2] is a multimodal dataset which includes video and introduces strategy annotations, both of which were missing in [1] (though they are different hidden role games). [2] even includes 8 games of *Avalon* as part of their test set for seeing how well models generalize across hidden role games.
* A larger issue is that the paper does not make an adequate appeal as to why this new dataset is needed (which could have helped address the novelty concern). Namely, the paper describes the benefits of the dataset as being significantly more challenging than the previous datasets due to increased game length. First, this claim does not seem to hold, since [2] reports 26,647 utterances across 199 games (8 of which are games of *Avalon*); this appears to be 134 utterances per game (not 64 as reported in the paper), which is longer than the 119 reported for the introduced dataset. Second, to adequately make a claim regarding increased complexity, the paper should detail experiments comparing to the previous datasets. Those datasets came out before the LLMs explored in this paper were released, so there are no adequate comparisons of performance to demonstrate the increased complexity of this dataset.

Addressing these issues could cause me to increase my soundess rating a bit:

* Table 3 must label the reported values as precision or accuracy. For all-role prediction, the caption makes it sound like the number before the slash should be avg precision and the number after is accuracy, but given the reported numbers that seems unlikely. I assume the column for *Evil find Merlin* is just accuracy?
* Why don't you report F1 score and instead rely on average precision? Both [1] and [2] report F1 scores.
* It's great that you performed multiple runs for the experiments, but how were the multiple runs and human annotations aggregated? Were they just averaged? If so, what was the variance or confidence interval? What was the agreement for the human eval?

[1] Samee Ibraheem, Gaoyue Zhou, and John DeNero. NAACL 2022. Putting the con in context: Identifying deceptive actors in the game of mafia.
[2] Bolin Lai, Hongxin Zhang, Miao Liu, Aryan Pariani, Fiona Ryan, Wenqi Jia, Shirley Anugrah Hayati, James M Rehg, and Diyi Yang. Findings of ACL 2023. Werewolf among us: A multimodal dataset for modeling persuasion behaviors in social deduction games.

**Reproducibility:**

4: Could mostly reproduce the results, but there may be some variation because of sample variance or minor variations in their interpretation of the protocol or method.

**Reviewer Confidence:**

5: Positive that my evaluation is correct. I read the paper very carefully and I am very familiar with related work.

**Typos Grammar Style And Presentation Improvements:**

* Terms like Macro AP should be written out explicitly so there is no confusion that it refers to avg precision (at least I assume that's what's being reported)

---

> ### Author Rebuttal · Authors · 2023-08-29
>
> We thank the reviewer for their insightful comments and suggestions provided in their review. We are glad to hear that the self-reported persuasion strategies from actual players have been perceived as favorable.
>
> **Highlights:**
> - We have added deception strategies (omission, commission, influence [5]) for each evil utterance that is identified as a lie.
> - We have added additional analysis with regard to the influence of player order and utilized lies on the model performance and outcome of the game
> - We added experiments utilizing the full game context with all models for better comparison of game representations and report our results as F1 scores (see large table below)
> - We have removed Vicuna in favor of Llama-2 and added GPT-3.5 and Llama-2 finetuning
> - We have added experiments across all models to predict persuasion strategies (see small table below)
>
> **Concern regarding novelty**: We would like to start by addressing the concern of novelty, particularly in comparison to [1] and [2]:
>
> **In comparison to Mafia [1],** our dataset does not allow for secret communication between the evil players; thus, synchronizing beliefs and strategies between players is more difficult. Further, we introduced a total of five different roles (Good, Merlin, Percival, Morgana, and evil) with different knowledge and objectives (while still being divided into the two general good/evil teams), unlike [1], which only considered the general good/evil role. Further, roles and team compositions in our dataset are fixed, lending themselves well to NLU tasks, while in Mafia, team sizes change throughout the game. Additionally, while [1] only focuses on predicting roles, we predict roles and persuasion strategies (we have added additional experiments, see small table below) as well as provide labels for the deception (i.e., lying) strategy of evil players (commission, omission, instigation [5]), which goes beyond the work done and even possible with [1] as the Mafia dataset does not have such labels. In our work, in response to other reviewers’ questions, we have added additional evaluations with regard to model performance and game outcome under various conditions, including the number of utterances per player type, the number and type of lies, and the position of Evil players and Merlin in the turn order
>
> **In comparison to Werewolf [2]:** Werewolf is a vastly different game than Avalon, as outlined in [2]. However, we acknowledge that [2] evaluates their approach on eight Avalon sourced from the Ego4D dataset. However, in Ego4D, these games do not have labeled player roles, game states or explicitly labeled deception strategies. As mentioned in [2] “_We [...] ask the annotators to look through each game clip and annotate the starting role, ending role, and the voting outcome of each player_”. Upon further investigation of this dataset, we find that the authors of [2] have not released the Avalon data, as they are “_still negotiating the Ego4D team about the data release_”, which, unfortunately, means that the data for the eight Avalon games (outside of the un-annotated videos from Ego4D) is unavailable, making our dataset the only available annotated Avalon dataset, containing five different roles, the full game state, eight labeled persuasion strategies, four labeled deception strategies, and a six-player focused chat that is completely task-related.
>
> **Summary of Novelty:**
> In our work, we propose a new benchmark dataset that complements yet goes beyond existing datasets on deception and persuasion. Avalon provides a setting in which current state-of-the-art LLMs struggle, due to the complexities inherent to deception, hidden objectives, and long-horizon dialogue between multiple players. Such problems are currently not studied in much detail due to the lack of sufficient datasets and benchmarks. We show that even when using state-of-the-art models and associated "tricks" (e.g., TypeChat [3] and structured beliefs), they still fail to match human performance. In particular, our proposed dataset is more challenging than existing datasets on similar games due to each game being longer, having more utterances, and having stricter rules that lend themselves well to NLU tasks. Further, our dataset is hand-labeled, largely by the players themselves, and contains various sets of labels including persuasion and deception strategies alongside the full game-state, and task-focused chat.
>
> **Calculation of average utterances:** We would also like to clarify the number of utterances in [2] and how we reached 68. Our games have an average of 119 utterances solely focusing on the game's content without any spurious utterances due to a naturalistic social setting. While [2] reports 26,647 utterances across 199 games, which results in 138 utterances per game, the authors of [2] also point out that “_49.2% of Ego4D utterances are labeled as no strategy because of the naturalistic social setting, while only 37.9% of YouTube utterances are labeled as no strategy since players from the YouTube videos are more proficient at the game and focused more on gameplay_”. This leads to the conclusion that 49.2% of Ego4D utterances are unrelated to the task at hand (because YouTube players were “_more proficient at the game and focused more on gameplay_”, which justifies the lower “_no strategy_” labels), namely the game of Avalon.
> Thus, we conclude that (1.0-0.492) * 26647 / 199 results in approximately 68 game-relevant utterances per game. As all utterances in our dataset are game-related, our dataset is more complex than the eight unreleased Avalon games provided in [2].
>
> **Detail experiments compared to the previous datasets:** We are happy to conduct additional experiments; however, since the Avalon data of [2] is not available, nor are their pre-trained models, we are unable to compare against their approach. However, we have provided additional experiments with our models on our dataset in order to predict our eight persuasion strategies, which are a superset of the six strategies in [2]. Results on the Micro-F1 score across all eight strategies can be found in the small table below. An extended version detailing the F1 scores for all individual classes has been added the the paper itself.
> With regards to [1], we would like to iterate again that the novelty of our work lies in the introduction of a new, complementary benchmark for long-horizon deception and persuasion in a cooperative-competitive multi-party setting. Statistically, we find Avalon to be a more difficult benchmark in terms of rules and game length as compared to [1], such that even SOTA LLMs can't perform well. Further, given the relatively small size of these rather complex datasets, we feel it's important to have multiple datasets that are capable of addressing such issues. Given that our dataset also comes with an online version of Avalon, we afford the ability to easily extend the dataset with little to no additional burden on the experimenters.
>
> In the following, we would like to address your questions:
>
> **Results in Table 3**: Regarding the reported scores in Table 3: We appreciate the comment on utilizing the F1 score and have updated our entire Table 3 (see large table below) accordingly over a total of 10 runs per model per modality and task. As the field of LLMs moved since initial submission, we have also included Llama-2 (in place of Vicuna due to better performance; however, Vicuna is now available in the supplemental material), while also adding fine-tuned versions of Llama-2-13b and GPT-3.5-turbo. Furthermore, we have added, for all models, results when utilizing the full context of each game (as much as possible, given their context lengths, truncating early rounds if necessary). In comparison, the round-based context utilizes an average of 974 (std: 333, max: 1941) tokens, while the full context utilizes an average of 2844 (std: 2011, max: 8556) tokens.
> In the old Table 3, the scores for Evil find Merlin were indeed just MultiClassAccuracy. However, we would also like to point out that we changed our methodology to utilizing TypeChat [3] (which we have re-implemented in Python), which makes it easier for the LLM to produce the structured output (and to correct it if it is non-conformant) relevant to our task; thus scores have slightly increased across the board.
>
> **Human Baseline**: For our six test games, we utilized three to four human annotators to read through each game’s chat log and game states in order to identify the roles. This is the same task and information given to LLMs in the Chat+State modality with full context history. Results were averaged over all annotators for each game. The agreement of the annotators measured as the mean of all pairwise Cohen Kappas is 0.41, indicating moderate agreement amongst the annotators according to [4].
>
> **Multiple persuasion labels:** The possibility of utilizing multiple persuasion strategies is indeed interesting. In our current setup, each utterance contains an average of 16 tokens, and participants were asked (not enforced) only to make “one argument per utterance message”, such that the need for multiple strategies in one chat message is reduced. This approach is also self-reinforcing during the labeling of their own utterances (approximately 82% of participants self-labeled their utterances). After choosing one of the eight possible strategies, no additional selections or changes can be made, thus weakly enforcing the idea of only placing one strategy in each utterance.
>
> **Why the spell checker:** Our game is set up in turns with a generous time limit for each turn in order to advance the game eventually and keep arguments reasonably concise. However, players rather type instead of correcting hastily misspelled words. Additionally, many of our participants were not native English speakers. Further, we use the spell checker with a custom dictionary (and high incentive) to correct the spelling of player names during the game. To address slang, we manually evaluated which words were corrected by the spell-checker and provided exceptions for slang like “hmmm” and similar terms. Special non-ASCII characters were not allowed; thus, emojis were not possible. However, emojis like “:)” have been left unaltered.
>
>
> **Additional Results:**
>
> **Prediction of Persuasion Strategies:**
> | PStrat | GPT-4 | GPT-3.5 |  GPT-3.5 (FT) | Llama-2-13b | Llama-2-13b (FT) |
> |----------|-------|---------|---------------|-------------|------------------|
> | Micro-F1 | 0.37  | 0.35    | 0.43          | 0.15        | 0.20             |
>
> **Updated Table 3 (Role Prediction)**
> In the table below, the first number represents the round-based inference with a carried-over belief, while the second number represents inference with the full context.
>
> | Model | Familiar with Avalon | Trained Model (14 Games) | Using Chat | Using State | Find Good | Find Evil | Find Merlin | Evil find Merlin (Final Guess) | Evil find Merlin (Anytime)  |
> | -- | -- | -- | -- | -- | -- | -- | -- | -- | -- |
> | Gpt-4      | [x] | | [x] | | 0.67 / 0.67 | 0.48 / 0.55 | 0.36 / 0.20 | 0.17 / 0.17  | 0.83 / 0.67  |
> | Gpt-4      | [x] | | | [x] | 0.59 / 0.57 | 0.20 / 0.33 | 0.06 / 0.31 | 0.17 / 0.33 | 0.17 / 0.50 |
> | Gpt-4      | [x] | | [x] | [x] | 0.67 / 0.68 | 0.46 / 0.58 | 0.05 / 0.27 | 0.00 / 0.00 | 0.67 / 0.50 |
> | gpt-3.5-turbo      | [x] | | [x] | |  0.68 / 0.60 | 0.46 / 0.40 | 0.23 / 0.17 | 0.17 / 0.17 | 0.33 / 0.50 |
> | gpt-3.5-turbo      | [x] | |  | [x] | 0.57 / 0.47 | 0.46 / 0.30 | 0.00 / 0.32 | 0.17 / 0.17 | 0.17 / 0.17 |
> | gpt-3.5-turbo      | [x] | | [x] | [x] | 0.58 / 0.65 | 0.34 / 0.47 | 0.23 / 0.13 | 0.17 / 0.17 | 0.33 / 0.33 |
> | gpt-3.5-turbo      | [x] | [x] | [x] | [x] | 0.52 / 0.59 | 0.38 / 0.41 | 0.19 / 0.15 | 0.17 / 0.17 | 1.00 / 0.67 |
> | Llama-2 |  | | [x] | | 0.68 / 0.61 | 0.39 / 0.27 | 0.00 / 0.00 | 0.17 / 0.00 | 0.17 / 0.17 |
> | Llama-2 |  | | | [x] | 0.41 / 0.62 | 0.00 / 0.34 | 0.00 / 0.00 | 0.00 / 0.00 | 0.00 / 0.17 |
> | Llama-2 |  | | [x] | [x] | 0.61 / 0.55 | 0.33 / 0.22 | 0.00 / 0.00 | 0.17 / 0.00 | 0.17 / 0.33 |
> | Llama-2 | [ ] | [x] | [x] | [x] | 0.65 / 0.63 | 0.35 / 0.26 | 0.23 / 0.27 | 0.33 / 0.00 | 0.33 / 0.00 |
> | Random |  |  |  |  | 0.50 | 0.34 | 0.17  | 0.17 | 0.60 |
> | Human | [x] | | [x] | [x] | 0.76 | 0.72 | 0.33 | 0.5 | 0.67  |
>
> 1. Samee Ibraheem, Gaoyue Zhou, and John DeNero. NAACL 2022. Putting the con in context: Identifying deceptive actors in the game of mafia
> 2. Bolin Lai, Hongxin Zhang, Miao Liu, Aryan Pariani, Fiona Ryan, Wenqi Jia, Shirley Anugrah Hayati, James M Rehg, and Diyi Yang. Findings of ACL 2023. Werewolf among us: A multimodal dataset for modeling persuasion behaviors in social deduction games.
> 3. Microsoft, TypeChat
> 4. Viera et al.: “Understanding interobserver agreement: the kappa statistic.”
> 5. The Three Types of Lies: Penn State University

---

### Official Review · Reviewer_YFpd · 2023-08-04

**Soundness:** 4

**Excitement:**

3: Ambivalent: It has merits (e.g., it reports state-of-the-art results, the idea is nice), but there are key weaknesses (e.g., it describes incremental work), and it can significantly benefit from another round of revision. However, I won't object to accepting it if my co-reviewers champion it.

**Paper Topic And Main Contributions:**

This paper presents a data collection of (massively) multi-party dialogue in an online game, in a cooperative / competitive scenario [new data resource]; a task defined on this data and some experiments showing LLM performance on it [NLP engineering experiment]


**Questions For The Authors:**

- As you do cite one paper from that group (Keizer et al 2017), you seem to be aware of the "Settlers of Catan" project. Would you not consider that a "high-quality dataset for a ling-horizon multi-party dialogue featuring deception and persuasion" (line 096)?


**Reasons To Accept:**

- Potentially very interesting resource that is interestingly different from more usual task-oriented dialogue (which is typically between two parties and purely cooperative) and also chit chat dialogue

- Experiments are well designed, executed, and evaluated / analysed

- Very well written


**Reasons To Reject:**

- Ultimately, it is a bit unclear to me what the take-home message is supposed to be. This clearly is a very hard task, that I would prima facie not have very strong reasons to think LLMs can handle. It does indeed turn out that they don't hande it very well. What have we learned?

- To elaborate on this, I think it would have been helpful to spend more time on analysing the data. Are there more circumscribed subtasks within the larger task "detect who's playing which role"? What subtypes of the language understanding capacity are targeted by this? What would an overhearer need to be able to do to solve this task? Spot contradictions? What are typical strategies that humans use to solve this task?


**Reproducibility:**

4: Could mostly reproduce the results, but there may be some variation because of sample variance or minor variations in their interpretation of the protocol or method.

**Reviewer Confidence:**

5: Positive that my evaluation is correct. I read the paper very carefully and I am very familiar with related work.

**Typos Grammar Style And Presentation Improvements:**

- line 156, something wrong with reference
- l 240, "i.e. the" --> "i.e.\ the"
- l 328, "the priviledged knowledge of the evil players are" --> "of who the evil players are"?
- l 412, "global sate" -> state
- l 460, "18 years old an familiar" -> and familiar

---

> ### Author Rebuttal · Authors · 2023-08-29
>
> We would like to thank the reviewer for their thorough review of our work and are encouraged to hear that our proposed dataset was perceived as an interesting resource that is different from the usual task-oriented dialogue. Furthermore, we are excited to hear that our experiments were perceived as well-designed, executed, evaluated, and analyzed.
>
> **Highlights:**
> - We have added deception strategies (omission, commission, influence [2]) for each evil utterance that is identified as a lie.
> - We have added additional analysis with regard to the influence of player order and utilized lies on the model performance and outcome of the game
> - We added experiments utilizing the full game context with all models for better comparison of game representations and report our results as F1 scores (see large table below)
> - We have removed Vicuna in favor of Llama-2 and added GPT-3.5 and Llama-2 finetuning.
> - We have added experiments across all models to predict persuasion strategies (see small table below)
>
> **What is the take-home message?:**
> We would like to elaborate on this point. Our analysis points to a more nuanced picture than simply that LLMs don’t handle it very well. While the LLM only outperforms the human in finding Merlin in one setting (GPT-4 using chat only), dramatically different performances can be observed compared to a random baseline. For example, LLMs tend to be capable of finding good players (as compared to a random baseline), while at the same time, struggling with identifying evil players and merlin correctly (similar to the random baseline). Further, we show that the utilization of lies on the side of evil players has a statistically significant impact on the model’s capability of identifying Merlin, i.e., the more lies evil people utilize, the more likely Merlin will be found out. Additional experiments to that end have been added to our paper and are outlined further below.
> Our take-home message is that Avalon provides a setting in which current state-of-the-art LLMs do poorly due to complex deception and long-horizon dialogue between multiple players, which is currently not studied in much detail due to the lack of sufficient datasets and benchmarks. We show that even when using state-of-the-art models and associated "tricks" (e.g., TypeChat [3] and structured beliefs), they still fail to match human performance. In particular, our proposed dataset is more challenging than existing datasets on similar games because each game is longer, has more utterances, and has stricter rules that lend themselves well to NLU tasks. For example, our corpus does not have changing roles and teams, nor secret communication. Furthermore, we consider our dataset to be more comprehensive by providing five different roles, the full game state (and show how it can be used), eight labeled persuasion strategies, four labeled deception strategies (different types of lies), and a six-player focused chat that is completely task-related. Finally, we conjecture that humans are prone to deception and that investigation in such matters is beneficial for the broader community,
>
> **Concern regarding lack of in-depth data analysis:**
> We appreciate the request for further analysis and have extended our paper as such. In particular, we have investigated the model’s performance and game outcome on: the number of utterances per player type, the number and type of lies, and the position of Evil players and Merlin in the turn order. We have found that Merlin is easier to identify when Evil has a high number of utterances being labeled as lies (particularly Commission), while it is more likely that Good will lose if Percival contributes a large number of utterances. Similarly, good players are easier to identify by our model if Percival or Morgana contribute many utterances, or if Merlin is early in the turn order.
>
> Further, to address the particular concerns of the reviewer:
>
> **Possible Additional Subtasks:** With our dataset, additional tasks are possible beyond role predicting, including developing agents that are advising players, agents that play the game themselves, identifying persuasion strategies (see results at the bottom), identification of deceptive strategies (we have added an additional set of labels to our dataset with the three most common lie-strategies: commission, omission, and influence) [3].
>
> **Learnable Language Capabilities:** With our dataset, we target long-horizon context learning in a challenging scenario utilizing hidden objectives, multi-party dialogue with many participants (introducing challenges when and how an LLM could enter a conversation), and further research on deception and persuasion in human communication. As humans are prone to deception, we conjecture that further research in that particular area is beneficial.
>
> **What would a Human Overhearer do?**: We have surveyed our human annotators again, asking them what cues they used to identify roles. As a qualitative summary, the answers were as follows: Evil players could be identified by their votes for or against parties, and if someone defends a person, that is likely to be evil. Merlin was identifiable by being more insightful than a normal player should be and by occasionally making “cover-up” claims that are vastly out of character. Generic good players were identified by exclusion through Evil/Merlin and generally had no direct strategy of being identified.
>
> **Human Strategies:** We have also asked a subset of our players what strategies they used during the game for each role. As a qualitative summary, good players should never lie while ensuring they do not contradict themselves without very good reason (i.e. when changing their mind about someone else). Evil players should not support their peers and wait for them to make a mistake, then call them out for it to make themselves look good. Finally, Merlin should be making a few wrong claims in cases in which the group already decided differently to hide as “being confused.”
>
> **“Settlers of Catan” being a high-quality dataset**: Thank you for asking for clarification on [1]. We consider the Settlers of Catan dataset a “high-quality dataset”; however, in their setting, the authors only focus on dialogue with artificial agents, not other humans, and limit the context of persuasive behavior to trading negotiations in a purely non-cooperative setting. Compared to the Settlers of Catan dataset, our proposed corpus introduces the following additional challenges: 1) long-horizon communication (avg. utterances in a game of Settlers: 79, ours: 119), 2) cooperative-competitive teamwork, 3) broader context of the discussion (i.e., not limited to individual trades), and 4) high-quality labels for persuasion and deception strategies. Further, Avalon highly depends on complex and sophisticated dialogue, while Settlers of Catan’s requirements are more forgiving as trades are handled independently. With these additional challenges and opportunities, we believe that our proposed dataset goes beyond the existing datasets on dialogue and persuasion, making it a relevant and exciting testbed for future research in agent development and investigation of deceptive behavior.
>
> **Additional Results:**
>
> **Prediction of Persuasion Strategies:**
> | PStrat | GPT-4 | GPT-3.5 |  GPT-3.5 (FT) | Llama-2-13b | Llama-2-13b (FT) |
> |----------|-------|---------|---------------|-------------|------------------|
> | Micro-F1 | 0.37  | 0.35    | 0.43          | 0.15        | 0.20             |
>
> **Updated Table 3 (Role Prediction)**
> In the table below, the first number represents the round-based inference with a carried-over belief, while the second number represents inference with the full context.
>
> | Model | Familiar with Avalon | Trained Model (14 Games) | Using Chat | Using State | Find Good | Find Evil | Find Merlin | Evil find Merlin (Final Guess) | Evil find Merlin (Anytime)  |
> | -- | -- | -- | -- | -- | -- | -- | -- | -- | -- |
> | Gpt-4      | [x] | | [x] | | 0.67 / 0.67 | 0.48 / 0.55 | 0.36 / 0.20 | 0.17 / 0.17  | 0.83 / 0.67  |
> | Gpt-4      | [x] | | | [x] | 0.59 / 0.57 | 0.20 / 0.33 | 0.06 / 0.31 | 0.17 / 0.33 | 0.17 / 0.50 |
> | Gpt-4      | [x] | | [x] | [x] | 0.67 / 0.68 | 0.46 / 0.58 | 0.05 / 0.27 | 0.00 / 0.00 | 0.67 / 0.50 |
> | gpt-3.5-turbo      | [x] | | [x] | |  0.68 / 0.60 | 0.46 / 0.40 | 0.23 / 0.17 | 0.17 / 0.17 | 0.33 / 0.50 |
> | gpt-3.5-turbo      | [x] | |  | [x] | 0.57 / 0.47 | 0.46 / 0.30 | 0.00 / 0.32 | 0.17 / 0.17 | 0.17 / 0.17 |
> | gpt-3.5-turbo      | [x] | | [x] | [x] | 0.58 / 0.65 | 0.34 / 0.47 | 0.23 / 0.13 | 0.17 / 0.17 | 0.33 / 0.33 |
> | gpt-3.5-turbo      | [x] | [x] | [x] | [x] | 0.52 / 0.59 | 0.38 / 0.41 | 0.19 / 0.15 | 0.17 / 0.17 | 1.00 / 0.67 |
> | Llama-2 |  | | [x] | | 0.68 / 0.61 | 0.39 / 0.27 | 0.00 / 0.00 | 0.17 / 0.00 | 0.17 / 0.17 |
> | Llama-2 |  | | | [x] | 0.41 / 0.62 | 0.00 / 0.34 | 0.00 / 0.00 | 0.00 / 0.00 | 0.00 / 0.17 |
> | Llama-2 |  | | [x] | [x] | 0.61 / 0.55 | 0.33 / 0.22 | 0.00 / 0.00 | 0.17 / 0.00 | 0.17 / 0.33 |
> | Llama-2 | [ ] | [x] | [x] | [x] | 0.65 / 0.63 | 0.35 / 0.26 | 0.23 / 0.27 | 0.33 / 0.00 | 0.33 / 0.00 |
> | Random |  |  |  |  | 0.50 | 0.34 | 0.17  | 0.17 | 0.60 |
> | Human | [x] | | [x] | [x] | 0.76 | 0.72 | 0.33 | 0.5 | 0.67  |
>
> 1. Kaizer et al.: “Evaluating Persuasion Strategies and Deep Reinforcement Learning
> Methods for Negotiation Dialogue Agents”
> 2. The Three Types of Lies: PennState University
> 3. Microsoft, TypeChat

---

### Meta-Review · Area_Chair_Vzrv · 2023-10-06

**Recommendation:** 5

**Metareview:**

This paper presents a potentially valuable resource, focusing on a unique domain of dialogue that differs from traditional task-oriented or chit-chat dialogues. The experiments are well-designed, executed, and thoroughly evaluated and analyzed. The paper is also well-written., Concerns raised by reviewers are adequately addressed in the rebuttal and author responses.

---

### Decision · Program_Chairs · 2023-10-07

**Decision:**

Accept-Findings

**Comment:**

This paper presents a potentially valuable resource, focusing on a unique domain of dialogue that differs from traditional task-oriented or chit-chat dialogues. The experiments are well-designed, executed, and thoroughly evaluated and analyzed. The paper is also well-written., Concerns raised by reviewers are adequately addressed in the rebuttal and author responses.